# Microplate Reader–TLC–HPLC–UPLC-MS: A Rapid Screening Strategy for Isoliquiritigenin-Transforming Bacteria

**DOI:** 10.3390/s25030827

**Published:** 2025-01-30

**Authors:** Chuanhong Nie, Ruiqi Liu, Songhao Yang, Panpan Li, Jing Zhang

**Affiliations:** State Key Laboratory Incubation Base for Conservation and Utilization of Bio-Resource in Tarim Basin & College of Life Science and Technology, Tarim University, Alar 843300, China; niechuanhong123@163.com (C.N.);

**Keywords:** MTHM method, isoliquiritigenin, transforming bacteria, large-scale screening, amentoflavone

## Abstract

This article primarily develops a new technology for the rapid large-scale screening of isoliquiritigenin-transforming strains based on the MTHM (microplate reader–TLC–HPLC–UPLC-MS) method. ISO is a chalcone compound with potential pharmacological activity, and its rich substitution sites on the benzene ring provide a solid foundation for structural modification and drug development. This study screened approximately 1500 strains and employed a microplate reader, thin-layer chromatography, high-performance liquid chromatography, and mass spectrometry to verify the transformation products, identifying 15 strains with significant transformation capabilities. This study demonstrates that the optimized MTHM method is efficient and reliable, capable of rapidly detecting subtle structural changes in flavonoids before and after microbial transformation. During the transformation process, bioactive flavonoid compounds, such as amentoflavone and 5′-methoxyflavonoid, were discovered. Additionally, the experiments revealed that Czapek medium, modified Martin medium, and LB medium exhibited high efficiency in screening transforming strains. This research provides new technical approaches for ISO structural optimization and drug development while highlighting the important application potential of microbial transformation in natural product development. Future studies could further explore the metabolic potential of these strains, optimize transformation conditions, and promote the application of ISO in the medical field.

## 1. Introduction

The roots and rhizomes of licorice have long been used as traditional Chinese medicine and natural additives [1], among which flavonoid compounds are polyphenolic compounds found in herbal medicines, and the important physiological functions of flavonoid compounds have increasingly attracted attention [2,3]. In addition, there are currently known to be 300 types of flavonoid compounds extracted from licorice [4], including flavonones [5], flavonols [6], chalcone [7], isoflavones [8], and isoflavans [9]. Chalcones are the most important class of secondary metabolites in licorice [10], not only serving as important synthetic precursors of flavonoids [11], but also representing a major form of natural products, exhibiting activities such as anti-inflammatory [12], antioxidant [13], antitumor [14], and antibacterial [15] activities. In addition, the main active flavonoid compounds in licorice include ISO, [16] which is the substance that we focus on.

ISO, a type of hydroxychalcone, contains three phenolic hydroxyl groups within its molecule, theoretically enhancing its polarity. However, in practical applications, this polarity poses a barrier to its transmembrane transport. Nevertheless, it is precisely the abundant substitution sites on its benzene ring that provide favorable conditions for scientists to enhance its biological activity through structural modification, indicating its broad prospects as a prodrug in the field of clinical medicine. Boyapelly [17] and his research team successfully transformed ISO into a prodrug form with significantly improved solubility (up to 9.6 mg/mL) through clever structural modification strategies, while maintaining excellent stability. Ma Yong ting [18] introduced 10 types of Mannich base groups, including dimethylamine, morpholine, and diisopropylamine, at the A ring 3′-methyl position of 3′-methyl ISO. In vivo experimental results showed that these derivatives exhibited better antitumor effects than the original ISO, especially the diisopropylamine derivative, which had the most significant inhibitory effect. Daye et al. [19,20] discovered that in the presence of chalcone synthase and chalcone reductase, three molecules of propanoyl-CoA and one molecule of coumaroyl-CoA react to form ISO, which is ultimately converted to butein after hydroxylation by chalcone-3-hydroxylase. By introducing different modifying groups, scientists also hope to further explore and enhance the pharmacological activity of ISO, contributing new strength to the field of drug development. Although a large body of evidence indicates that ISO can inhibit the growth of certain cancers by regulating specific signaling pathways, there remains a significant gap between basic research and clinical application due to its poor bioavailability and delivery characteristics [3,21].

Most studies on isoflavonoids and flavonoid compounds have focused on plant sources, while there has been relatively little attention on microorganisms as potential sources of these compounds [22]. The existence of metabolic pathways for the biosynthesis of isoflavonoids and flavonoids in microorganisms has been confirmed. In particular, the presence of isoflavonoids and flavonoid compounds in fungi has been widely reported in research [23]. Fungi exhibit unique and unusual biochemical pathways, and many important drugs are derived from their secondary metabolites, such as penicillin, cyclosporin, paclitaxel, and statins [24]. Biotransformation is an alternative strategy with great potential for producing novel bioactive flavonoid compounds [25]. Since 2010, several qualitative reviews discussing the biotransformation of flavonoids by microorganisms have been published, highlighting the prospects of microbial biotransformation as a cost-effective and environmentally friendly drug design tool [26]. The application of microorganisms in the biotransformation of chalcone has led to the formation of many novel flavonoid compounds through cyclization, hydroxylation, reduction, demethylation, and dehydrogenation reactions in various mixtures [27,28]. For example, *Aspergillus saitoi* hydroxylates 4-Hydroxyderricin at the C-3 position of ring B, forming 3,4-dihydroxyderricin [29]. Sanchez-Gonzalez and Rosazza found that *Aspergillus alliaceus* UI 315 demethylates the C-3 methoxy of 2′-hydroxy-2,3-dimethoxychalcone to produce 2′,3-dihydroxy-2-methoxychalcone. *Aspergillus alliaceus* UI 315 can cyclize 2′-hydroxy-2,3-dimethoxychalcone to form 2′,3′-dimethoxyflavonone and further demethylate the C-3′ methoxy to yield 3′-hydroxy-2′-dimethoxyflavonone [30]. Hydrogenation of flavonoid compounds has only been reported during the biotransformation of chalcone to dihydrochalcone. *Gordonia* sp. DSM 44456 and *Rhodococcus* sp. DSM 364 can hydrogenate chalcone and 4-methoxychalcone to dihydroxychalcone, and *Rhodococcus* sp. DSM 364 hydrogenates chalcone aglycone to orobol [31]. *Paspalum maritimum* Trin. Flavus converts hydrogenated chalcone to dihydrochalcone [32]. *Aspergillus niger* can selectively cyclize chalcone to flavonone, which simulates the plant biosynthetic process. For example, 2′-hydroxy-2″,4″-dimethoxychalcone is transformed into 2′,4′-dimethoxyflavonone, 2′-methoxy-4′-hydroxyflavonone, and 2′-methoxy-8,4′-dihydroxyflavonone. This transformation first involves enzyme-catalyzed cyclization and demethylation, followed by hydroxylation [33] (Figure 1). In summary, the cyclization of isoprenylflavonone and chalcone has been widely reported. Isoprenylflavonone cyclizes at the isoprenyl side chain, forming a new five-membered ring connected to the A ring, while the chalcone region selectively cyclizes to flavonone [34]. *Cunninghamella, Penicillium*, and *Aspergillus* strains are very capable of transforming flavonoid compounds and can almost perform all reactions [22].

A microplate reader is used to measure the uptake of protein-loaded nanoparticles by cells by detecting the fluorescence intensity related to the cells [35]. For example, Caco-2/mucin or Caco-2/HT-29 co-cultures can be treated with fluorescently labeled nanoparticles for 3 h, followed by thorough washing to eliminate the secreted mucus adhering to the surface of the samples [36]. A microplate reader is used to examine the effects of mucus on the internalization of nanoparticles and the barrier internal formulation dose [37]. Additionally, a microplate reader is used as a tool for real-time gene expression analysis based on imaging with biosensors of transcription factors in live cells grown on a solid culture medium [38]. The principle of the microplate reader is that the absorbance of a substance at a certain wavelength is linearly related to its concentration, and the characteristics of the microplate reader are reflected in its ability to measure a large number of samples, generally 96, with fast measurement speed, simple operation, low reagent consumption, and short time, greatly improving work efficiency. However, due to the small sample capacity and instrument structure, the MTHM method has not been used for the screening of transformation strains. Our preliminary experiments found that flavonoid compounds have ultraviolet absorption in the 200–400 nm range, which is within the detection range of a microplate reader, and a microplate reader can distinguish subtle changes in flavonoid structures before and after microbial transformation, which has been experimentally proven. Therefore, the MTHM method was adopted for large-scale screening of transformation strains.

In summary, isoflavone has multiple modification sites and the advantages of microbial transformation (directional, rapid reaction, non-toxic), but currently, there is a lack of methods for screening isoflavone transformation strains, and large-scale screening of the MTHM method has not been conducted. This study establishes a method for rapid large-scale screening of isoflavone transformation strains using a microplate reader as the detection instrument. Reagents and samples are added to react under set conditions, and the optical density (OD) value of the 96-well plate is measured with the microplate reader. The screening results from the microplate reader are finally verified using thin-layer chromatography (TLC), high-performance liquid chromatography (HPLC), and ultra-high-performance liquid chromatography–mass spectrometry (UPLC-MS) (Figure 2). The MTHM method proposed in this paper provides a new approach to microbial drug discovery and offers new ideas for finding new flavonoids.

## 2. Materials and Methods

### 2.1. Materials and Reagents

The materials used were as follows: ISO, CAS number: 961-29-5, Shanghai Yuan Ye Biotechnology Co., Ltd (Shanghai, China); Trypticase Soy Broth (TSB), Czapek-Dox Agar (CMM), R_2_A Medium, International Streptomyces Project Medium 4 Medium (ISP4), Medium Luria–Bertani Medium (LB), Potato Dextrose Agar Medium (PDA), Modified Martin’s Medium (MMM), Gao’s No. 1 (GS 1), Medium and Nutrient Agar Medium (NA), methanol, acetonitrile, formic acid, ethyl acetate, petroleum ether. SW-CJ-2F clean bench, Shanghai Boxun Medical Biological Instrument Co., Ltd. (Shanghai, China); Epoch full-wavelength microplate reader, BioTek Instruments, Inc.; LDZP5OL-1 Vertical High-Pressure Steam Sterilizer, Shanghai Shen’an Medical Equipment Factory; DHG-9070A Electric Constant Temperature Forced Air Drying Oven, Shanghai Qixin Scientific Instrument Co., Ltd.; SL-PL+ Pipette, Mettler Toledo Technology (Shanghai, China) Co., Ltd.; DHP-9082 Electric Constant Temperature Incubator, Shanghai Yiheng Scientific Instrument Co., Ltd. (Shanghai, China); HT150R High-Speed Tabletop Refrigerated Centrifuge, Hunan Xiangyi Laboratory Instrument Development Co., Ltd. (Changsha, China); BCD-272WDPD Refrigerator, High-Performance Liquid Chromatograph, Japan Shimadzu Corporation (Kyoto, Japan).

### 2.2. Experimental Methods

#### 2.2.1. Sample Collection, Strain Isolation and Identification

Sample collection: In the early stage of the laboratory, soil samples from licorice residue, licorice root, and Altun Mountain were collected from Xinjiang Alar New Agricultural Licorice Industry Co., Ltd. (Aksu City, China) for strain isolation.

Strain isolation: Using a sterile scalpel, we cut the surface-sterilized licorice root and stem into small pieces of 0.5 cm in length and cut the leaves into thin slices of 0.5 cm × 0.5 cm. We placed these samples separately onto the surfaces of the following media: TSB, CMM, R_2_A, ISP4, LB, PDA, MMM, GS 1, NA. We incubated these samples at a constant temperature (fungi at 28 °C; bacteria at 37 °C) to allow colony formation. We performed streak purification to isolate pure strains and stored the purified strains in 40% glycerol tubes for future preservation and subsequent identification.

Strain identification: The genomic DNA of the fungi was extracted using a modified CTAB method, amplified using ITS1 and ITS4 primers, and the amplified products were sequenced. The obtained 16S rDNA sequences were submitted to NCBI Blast (https://blast.ncbi.nlm.nih.gov/Blast.cgi) (accessed on 25 September 2024) for similarity comparison search, and the similarity of the strains was analyzed using the NCBI BLAST (basic local alignment search tool). Further, a phylogenetic tree was constructed using MEGA 7 software for phylogenetic analysis of the strains.

#### 2.2.2. Microplate Reader Methodological Investigation

Specificity: We took 200 μL of methanol, isoflavone solution, TSB culture medium, and Chai’s medium to the microplate, performed detection under the microplate reader, and recorded the OD value.

Linear relationship: We dissolved 1 mg of isoflavone in a 10 mL volumetric flask, added methanol to the mark, and shook the mixture well to obtain a stock solution with a mass concentration of 100 μg/mL. We added 200 μL of the above stock solution to the microplate using a serial dilution method and added methanol solution to each well to make a total volume of 200 μL, measuring the OD value at wavelengths of 200–500 nm in the microplate reader. We diluted the reference substance 20 times and 100 times, using the same method to detect its OD value.

The detection limit was calculated using the formula LOD = 3.3 δ/S, where δ is the standard deviation of the blank sample and S is the slope of the standard curve.

The quantification limit was calculated using the formula LOQ = 10 δ/S, where δ is the standard deviation of the blank sample and S is the slope of the standard curve.

Reproducibility: We selected the isoflavone standard, precisely weighed 1 mg six times, diluted it to 10 mL in a volumetric flask, and performed detection under the microplate reader, recording its OD value. Relative standard deviation RSD = S/$\bar{x}$, where S is the sample standard deviation and $\bar{x}$ is the mean of the sample dataset. 

#### 2.2.3. Determination of Different Types of Culture Media Added with Isoflavone

We selected 100 μL of TSB, CMM, R_2_A, ISP4, PDA, GS 1, NA, MMM, and LB culture media for mixing with 100 μL of isoflavone at a concentration of 50 μg/mL in a 96-well plate. We detected the OD value using a microplate reader with a wavelength range of 200–500 nm and observed the recorded results.

#### 2.2.4. Test of the Microplate Reader’s Sensitivity to Slight Changes in Flavonoid Structure

The concentration of 100 μg/mL ISO and chalcone B, licorice chalcone, liquiritigenin, neoisoliquiritin, isoliquiritin, glabridin, soybean isoflavone, liquiritin, glabridin, licorice phenol, and licorice phenol were prepared. Five gradients were set up in 96-well plates, with a total volume of 200 μL. Other flavonoid standard solutions, 100 μL, 80 μL, 50 μL, 40 μL, 20 μL, were added in turn. Less than 200 μL was supplemented with methanol solution, and the results were observed.

#### 2.2.5. The Effect of Time on the Strain and the Repeatability of the Strain

The purified strains TRM19827 and TRM19829 and ISO solution were mixed and placed in a 20 mL PA bottle for fermentation culture. One strain was repeated five times and detected under a microplate reader to observe the results.

#### 2.2.6. Large-Scale Strain Screening of Transformants

About 1500 strains of Altun Mountain scientific research strains were activated with TSB medium and detected under a microplate reader, and the results were observed.

#### 2.2.7. Thin-Layer Chromatography Method Validation

Sample application was performed on a silica gel plate for thin-layer chromatography. Using the isoflavone standard substance (50 μg/mL), the developing agent ratio was methanol/ethyl acetate/petroleum ether = 1:10:16, and the results were observed under a UV analyzer in a dark box.

#### 2.2.8. High-Performance Liquid Chromatography Validation

The chromatographic column was a ZORBAX Eclipse Plus C18 (250 nm × 4.6 nm, 5 μm). The gradient elution procedure was as follows: 0–5 min, 10% B; 5–8 min, 15% B; 8–10 min, 50% B; 10–18 min, 70% B; 18–28 min, 85% B; 28–32 min, 100% B; 32–40 min, 5% B; The mobile phase was 0.1% formic acid water (solvent A) and acetonitrile/methanol = 9:1 (solvent B), the flow rate was 1.0 mL/min, the injection volume was 10 μL, and the detection wavelength was 367 nm.

### 2.3. Ultra-High-Performance Liquid Chromatography–Mass Spectrometry Validation

The fermentation broths of strains TRM19827 and TRM19829 were mixed with ISO for 48 h and detected by ultra-high-performance liquid chromatography–mass spectrometry. The time program was consistent with the time program of high-performance liquid chromatography. The mobile phase A was water, the mobile phase B was methanol, and the injection volume was 10 μL.

## 3. Results

### 3.1. Results of Strain Isolation

To rapidly identify isoflavone transformation strains, we used LB culture medium, R_2_A medium, CMM medium, ISP4 medium, NA medium, PDA medium, MMM medium, GS 1 medium, TSB medium, and samples from licorice roots, licorice residues, and Arjinshan soil to isolate a total of 122 strains using the dilution plating method and streak plating method (Table 1).

As can be seen from the table, a total of 122 strains were obtained from three samples across ten culture media. The probability of obtaining transformation strains from Chai’s medium, LB culture medium, and improved Martin’s medium was relatively high, with the probability of obtaining transformation strains from Chai’s medium being 31.43%, from LB culture medium being 14.29%, and from improved Martin’s medium being 20%.

### 3.2. Investigation of Microplate Reader Methodology

In order to verify whether the microplate reader has methodological capability for the isoflavone solution, we conducted some methodological investigations, and the results met our expectations.

#### 3.2.1. Specificity

Take methanol solution, isoflavone solution (100 μg/mL), TSB culture medium, and Chai’s medium 200 μL for detection under the microplate reader (Figure 3A). As can be seen from the figure, the culture medium and methanol reagent do not affect the detection of the isoflavone solution under the microplate reader.

#### 3.2.2. Investigation of Linear Relationship

Initial selection concentration range: Prepare isoflavone 500 μg/mL, then perform serial dilution seven times, using blank methanol as a control, and perform detection under the microplate reader (Figure 3B). As shown in the figure, the detection of isoflavone concentration by the microplate reader has limitations; when the concentration exceeds 125 μg/mL, the sensitivity of the microplate reader is very low, which is not conducive to the detection of isoflavone. Moreover, it was found that the maximum ultraviolet absorption wavelength of isoflavone under the microplate reader is 367 nm.

The relationship between the selected concentration and OD: Prepare Isoflavone at three concentration ranges of 5–100 μg/mL, 0.5–5 μg/mL, and 0.05–0.5 μg/mL, and perform detection under the microplate reader (Figure 3C–H). As shown in the figure, the concentration of isoflavone exhibits a good linear relationship at 5–100 μg/mL with wavelengths 292–351 nm and 385–448 nm. The concentration at 0.5–5 μg/mL shows a good linear relationship at wavelengths 328–405 nm, while the concentration of 0.05–0.5 μg/mL exhibits a poor linear relationship under the microplate reader.

#### 3.2.3. Reproducibility, Detection Limit, and Quantification Limit Investigation

Six 50 μg/mL ISO solutions were prepared and measured under a microplate reader. Based on the table (Table 2) and the above experiments, the detection limit and quantification limit of isoflavone were calculated using the slope of the standard curve. The detection limit of isoflavone was found to be 0.017 μg/mL, and the quantification limit was 0.05 μg/mL. From the table, the relative standard deviation of isoflavone was calculated to be 0.194%, indicating good reproducibility of isoflavone.

### 3.3. The Effect of Different Culture Media on Isoflavone Solution

Prepare a standard solution of Isoflavone at 50 μg/mL, using the microplate strip method for serial dilution of isoflavone from 200 μL to 0 μL, and add culture medium solution to 200 μL, then perform detection with the microplate reader. As shown in the figure (Figure 4), Chai’s medium, TSB culture medium, LB culture medium, Improved Martin’s medium, and NA medium have little effect on the isoflavone solution, while other media have a significant impact, which is detrimental to the detection of the isoflavone solution.

### 3.4. Gradient Mixing of Isoflavone Analogs and Mixing with Strains

This experiment investigates whether ISO can be detected under a microplate reader after mixing with other standards. The preparation included a 100 μg/mL concentration of ISO and Chalcone B, Glycyrrhizic Chalcone, Liquiritin, Neobavaisoflavone, Isoflavone Glycoside, Glabridin, Soy Isoflavones, Glycyrrhizin, Liquiritigenin, Glycyrrhetinic Acid in 96-well plates set with five gradients, with a total volume of 200 μL, sequentially adding Chalcone B solution 100 μL, 80 μL, 50 μL, 40 μL, and 20 μL and making up to 200 μL with ISO solution. We can see from the figure that when isoflavone is mixed with Soy Isoflavones, significant differences are observed under the microplate reader (Figure 5). The differences in OD values facilitate our subsequent selection of a difference to screen transformation strains, providing substantial assistance for our future experiments. It can also be observed that compounds with similar structures do not exhibit large differences.

### 3.5. The Results of the Effect of Time on the Strain and the Repeatability of the Strain

In order to investigate the effect of time on the strain and the repeatability of the strain, as well as the error caused by the microplate reader itself, we made five replicates of one strain and made two empty methanol controls to obtain a mean value. It was found that the OD difference of the strain was not more than 0.1, and the OD value of methanol was also less than 0.1, indicating that the strain itself and the microplate reader itself had little effect on the ISO solution.

### 3.6. Large-Scale Screening of Transformants

To demonstrate the reliability of the method, we verified it using approximately 1500 strains isolated from the laboratory and the A’erjin Mountain scientific expedition strains. We mixed the purified bacterial fermentation broth with ISO solution, cultured it for 48 h, and then detected it with a microplate reader. We used about 1500 strains to test the proposed MTHL method, and the results are promising (Figure 6). It was found that the OD values after 48 h would be lower than the OD values at 0 h. A very small number of strains had OD values higher than at 0 h, which is also consistent with our experiments. ISO is transformed, and its ultraviolet absorption under the microplate reader is reduced, leading to a decrease in OD values. We tested approximately 1500 strains using the proposed MTHM method, and the results are promising (Figure 6). Fifteen reliable strains were selected, namely TRM19771 (*Bacillus massiliosenegalensis*), TRM19773 (*Penicillium polonicum*), TRM19778 (*Bacillus* sp.), TRM19779 (*Penicillium* sp.), TRM19780 (*Penicillium* sp.), TRM19782 (*Penicillium mononematosu*), TRM19795 (*Penicillium mononematosum*), TRM19796 (*Penicillium roqueforti*), TRM19797 (*Penicillium paneum*), TRM19798 (*Penicillium roqueforti*), TRM19800 (*Fusarium solani*), TRM19801 (*Doratomyces* sp.), TRM19802 (*Fusarium solani)*, TRM19827 (*Penicillium* sp.), and TRM19829 (*Penicillium crustosum*). Among these, nine strains belong to the *Penicillium* genus, two strains belong to the Bacillus genus, two strains belong to the *Fusarium* genus, one strain belongs to the *Aspergillus* genus, and one strain belongs to the *Cephalosporium* genus. We also established the phylogenetic tree of these strains (Figure 7) and wanted to predict more strains with transformation ability through the classification status of the strains, such as *Bacillus*. This experiment verifies the strong transformation ability of fungi, which is consistent with the literature reported in our preface. Most bacteria do not have the ability to transform ISO.

### 3.7. Thin-Layer Chromatography Verification Results

Transfer 1 mL of the mixture of isoflavone and bacterial liquid to a centrifuge tube, prepare a developing agent mixture of methanol, ethyl acetate, and petroleum ether 20 mL, and place it in the developing tank for 15–20 min, with the ratio of developing agents being methanol/ethyl acetate/petroleum ether = 1:10:16, and observe in the dark box of the ultraviolet spectrophotometer at 254 nm (Figure 8A,B). As can be seen from the figure, the mixture of bacterial liquid and isoflavone solution exhibits a different position on the thin-layer plate compared to the isoflavone solution, and no other substances were observed after mixing the culture medium solution with the isoflavone solution, indicating that the mixture produced other substances, and the culture medium did not interfere with the experiment, providing a basis for subsequent experiments.

### 3.8. High-Performance Liquid Chromatography Results

After taking the culture 48 h, the fermentation broth of the strain was mixed with 1 mL of the isoflavone solution and filtered through a sterile 0.22 μm microporous filter membrane into a liquid phase vial, followed by detection using high-performance liquid chromatography (Figure 8C,D). As can be seen from the figure, there are no peaks at the position of the standard substance in the control group bacterial solution and culture medium, indicating that it does not affect the experimental results. Comparing the fermentation broth of the strain and the isoflavone solution mixture at 0 h, the mixture at 48 h shows a peak next to the standard substance, indicating that there are substances in the mixture that convert the standard substance into other substances.

### 3.9. Results of Liquid Chromatography–Mass Spectrometry

The fermentation broth of strains TRM19827 and TRM19829 was mixed with ISO for 48 h and detected by liquid chromatography–mass spectrometry. Some flavonoids were found according to the chromatographic response values of liquid chromatography–mass spectrometry (Table 3). The results showed that ISO was not found, but other active flavonoid analogs were found, such as amentoflavones, 5′-methoxyflavones, and other flavonoids. We also found two compounds with the same UPLC-MS chromatographic results on the map (Figure 9).

## 4. Discussion

This study systematically explored isoflavone and its biotransformation, particularly achieving significant findings in optimizing screening conditions for the microplate reader and studying microbial transformation efficiency. The results indicate that the biotransformation of isoflavone can be achieved not only through microorganisms but also significantly improved in conversion efficiency and product diversity by optimizing the culture medium and color reagent selection.

The strain screening part showed the separation efficiency of different media. Czapek’s medium, modified Martin medium. and LB medium had significant advantages in improving the screening efficiency of target strains, and 31.43%, 20%, and 14.29% of the transformed strains were screened, respectively. This is related to the nutritional components and environmental conditions of these three media. The nutrient composition of the medium promotes the rapid growth of microorganisms, thereby improving the conversion efficiency. In the subsequent determination of the microplate reader, the Czapek medium, the modified Martin medium, and the LB medium also showed a small effect on the ISO solution, which was also consistent with the experimental use of the microplate reader to screen the flavonoid-transforming bacteria.

In the methodology of detection using the microplate reader, this study clarified the detection linear range and detection limit of isoflavone. The results showed that within the concentration range of 5–100 μg/mL, the OD value of isoflavone exhibited a good linear relationship with concentration, and the detection limit was 0.017 μg/mL. This not only provides a reliable method for the quantitative analysis of isoflavone but also lays the foundation for subsequent high-throughput screening. More importantly, the strains selected through the microplate reader exhibited high conversion activity in both thin-layer chromatography and high-performance liquid chromatography validation, indicating the reliability and sensitivity of the microplate reader screening. This experimental design was not mentioned in previous articles. By detecting flavonoid compounds using the microplate reader, target strains were rapidly screened, and the selected target strains were ultimately validated through thin-layer chromatography, high-performance liquid chromatography, and liquid chromatography–mass spectrometry combined detection.

Regarding the structural modification of chalcone compounds, previous studies have indicated that phenylpropanoids and polyketone compounds are typically generated into chalcone under the catalysis of chalcone synthase, which then cyclizes to form flavonoid compounds. Isoflavonoid compounds share the same biochemical pathway as flavonoid compounds, derived under the catalysis of 2-hydroxyisoflavanone synthase through aryl migration in the 2-phenylbenzodihydropyran skeleton [39,40,41,42]. For the transformation of iso liquiritin, this study further explored the transformation products after mixing the strains with iso liquiritin. Through high-performance liquid chromatography and liquid chromatography–mass spectrometry analysis, it was found that the strain could transform isoflavone into various flavonoid compounds with potential bioactivity, such as 6-methoxy-2-(2-phenylethyl) chromone. Wang and Li et al. [43,44] isolated this compound from A. sinensis and demonstrated good inhibitory activity by inhibiting the production of superoxide and the release of elastase from neutrophils, with the expectation of developing it as a potential candidate for the treatment or prevention of various inflammatory diseases. Xiong et al. [45] described the bioactivity of amentoflavone from multiple aspects, and AMF improves inflammation by inhibiting the NF-KB signaling pathway and the activation of downstream target genes [46]. AMF protects the nervous system and prevents bone diseases due to its antioxidant and anti-inflammatory activities [47]. In addition, AMF can restore the imbalance of lipid and carbohydrate metabolism and reverse DNA damage caused by radiation [48]. AMF increases the expression of apoptosis and autophagy-related proteins, inhibits the expression of cell-cycle- and metastasis-related proteins, and leads to the control of cancer progression [48,49]. The flavonoid glycosides extracted by Priyanka Sati et al. [50] are structurally similar to 5′-methoxy ginkgo flavonoids and also exhibit high antibacterial and antioxidant activities. These products have significant research value in the fields of antioxidant, anti-inflammatory, antibacterial, and anticancer studies, providing new possibilities for the application prospects of isoflavone.

This study also has some limitations. For example, the specific mechanisms of the structural modification of isoflavone on its pharmacological activity are not yet clear and need to be elucidated through more molecular biology and pharmacology research. In addition, the identification of metabolic pathways and key enzymes in the microbial transformation process is also an important direction for future research. By conducting in-depth studies on these topics, more theoretical basis and technical support can be provided for the structural optimization and pharmacological activity enhancement of isoflavone.

Overall, this study provides new insights into the biotransformation of isoflavone, particularly in the screening of efficient strains and the optimization of transformation conditions, which is of significant importance. In the future, the potential of these strains can be further explored by combining bioinformatics and metabolomics, providing a solid foundation for the development and application of natural products.

## 5. Conclusions

This study successfully developed an efficient technique for the rapid screening of isoliquiritigenin-transforming strains using the MTHM (microplate reader–TLC–HPLC–UPLC-MS) method. By screening approximately 1500 strains with various culture media, it was found that CMM, MMM, and LB significantly improved the efficiency of selecting transforming strains. Ultimately, 15 strains with the ability to transform ISO were identified, with Penicillium accounting for 60%, demonstrating strong transformation potential. This study showed that this method could rapidly detect microbial transformation products of ISO such as amentoflavone and 5′-methoxyflavonoid, which possess potential biological activity. This provides not only new insights into the structural optimization and drug development of ISO but also a novel approach to discovering new flavonoid compounds through microbial transformation technology.

## Figures and Tables

**Figure 1 sensors-25-00827-f001:**
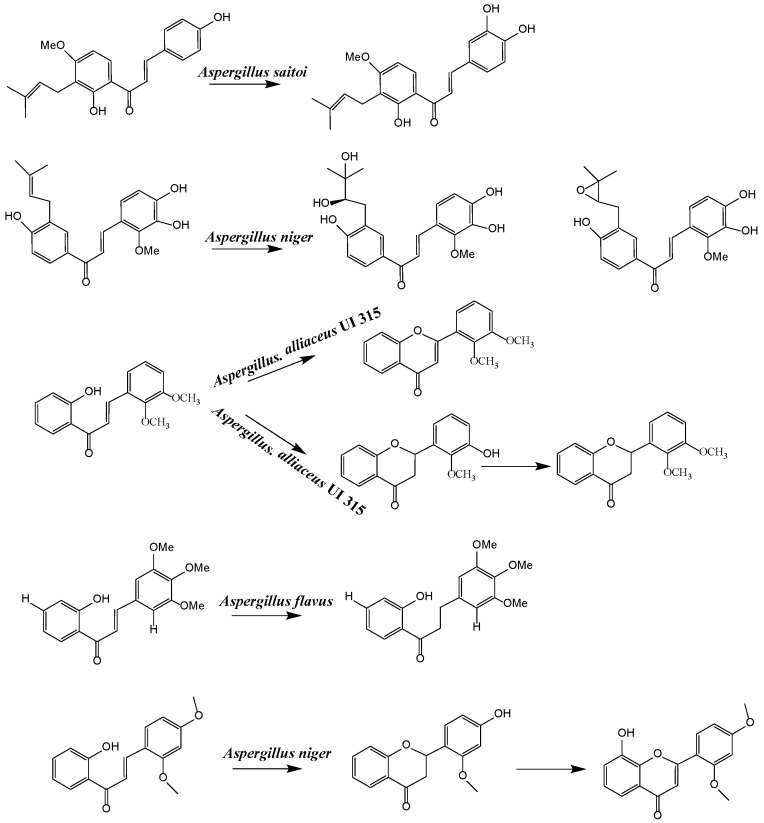
Strains transform chalcone structure.

**Figure 2 sensors-25-00827-f002:**
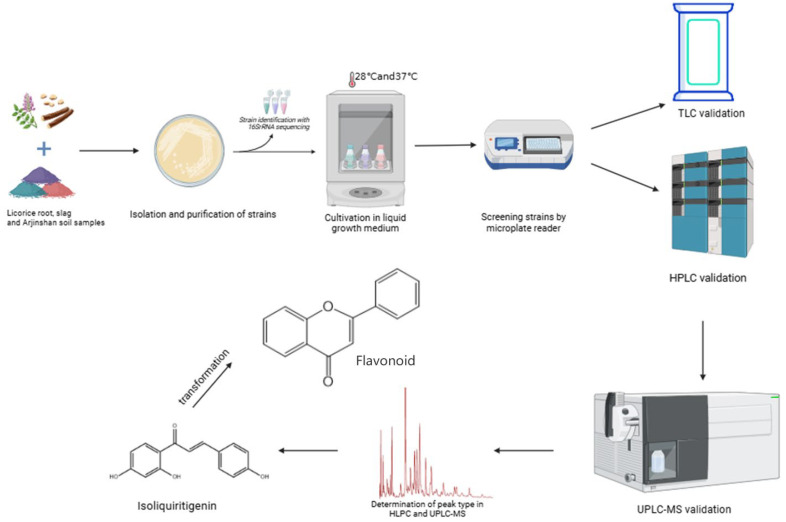
Technology roadmap.

**Figure 3 sensors-25-00827-f003:**
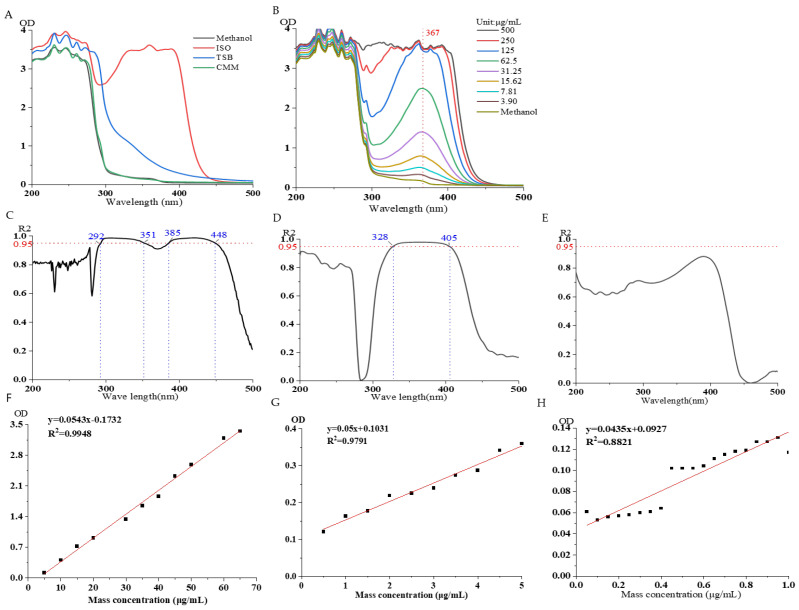
Investigation of specificity and linear relationship of ISO microplate reader. (**A**) Specificity of ISO. (**B**) Linear relationship of ISO. (**C**) The relationship between ISO concentration (5–100 μg/mL) and R2 at wavelength. (**D**) The relationship between ISO concentration (0.5–5 μg/mL) and R2 at wavelength. (**E**) The relationship between ISO concentration (0.05–0.5 μg/mL) and R2 at wavelength. (**F**) The relationship between ISO concentration (5–70 μg/mL) and OD value at 367 nm. (**G**) The relationship between ISO concentration (0.5–5 μg/mL) and OD value at 367 nm. (**H**) The relationship between ISO concentration (0.05–0.5 μg/mL) and OD value at 367 nm.

**Figure 4 sensors-25-00827-f004:**
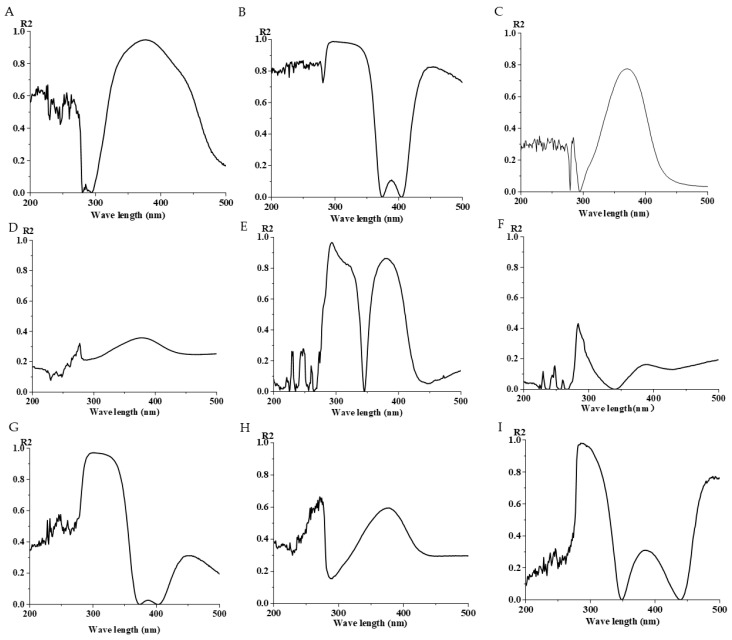
Differences between different media and ISO mixture. (**A**) The effect of CMM on ISO solution. (**B**) The effect of TSB culture medium on ISO solution. (**C**) The effect of R_2_A culture medium on ISO solution. (**D**) The effect of ISP4 culture medium on ISO solution. (**E**) The effect of LB culture medium on ISO solution. (**F**) The effect of PDA culture medium on ISO solution. (**G**) The effect of improved MMM on ISO solution. (**H**) The effect of GS 1 medium on ISO solution. (**I**) The effect of NA culture medium on ISO solution.

**Figure 5 sensors-25-00827-f005:**
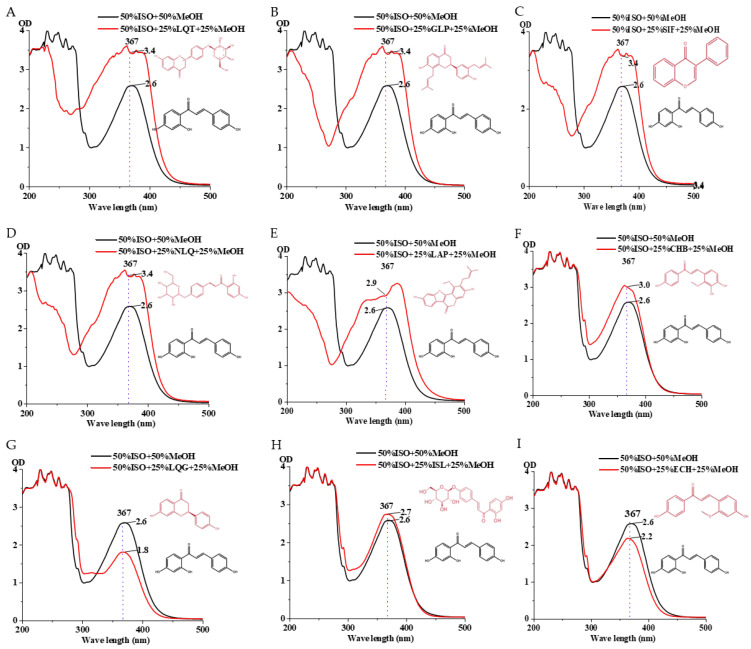
The mixture of ISO and flavonoid compounds, with ISO concentration at 50%, other flavonoid compounds at 25%, and methanol concentration at 25%. (**A**) LQT + ISO, (**B**) GLP + ISO, (**C**) SIF + ISO, (**D**) NLQ + ISO, (**E**) LAP + ISO, (**F**) CHB + ISO, (**G**) LQG + ISO, (**H**) ISL + ISO, (**I**) ECH + ISO.

**Figure 6 sensors-25-00827-f006:**
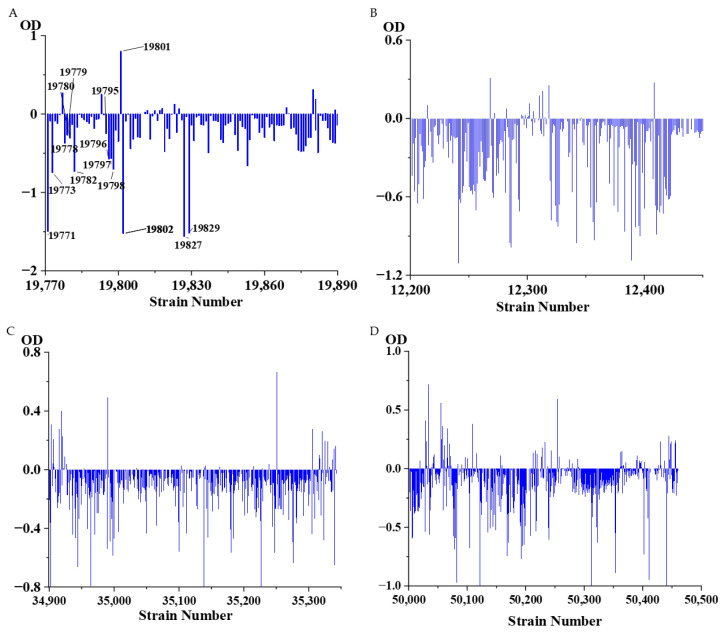
Strains and ISO mixed 0 h and 48 h OD difference. (**A**) Strain No. 19770-19890. (**B**) Strain No. 12200-12412. (**C**) Strain No. 34900-35342. (**D**) Strain No. 50000-50460. The OD value represents the difference in OD between the fermentation broth of the strains after 48 h and the isoflavone culture at 48 and 0 h.

**Figure 7 sensors-25-00827-f007:**
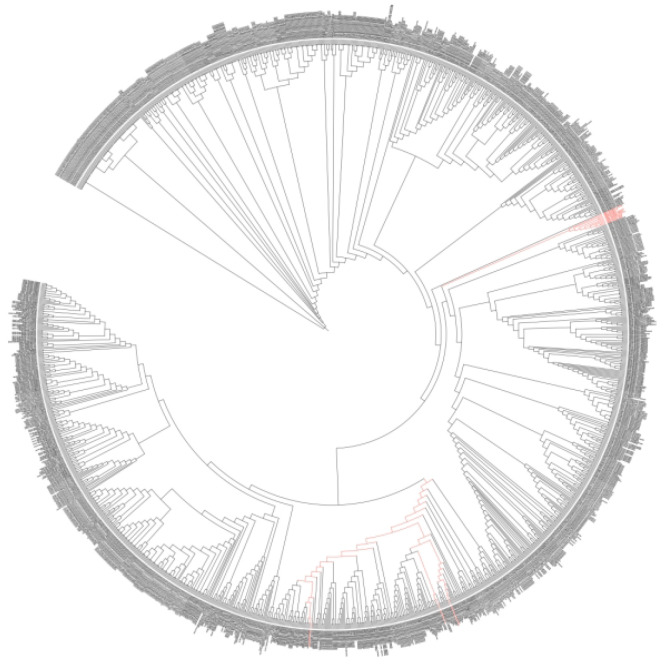
Phylogenetic tree of strains.

**Figure 8 sensors-25-00827-f008:**
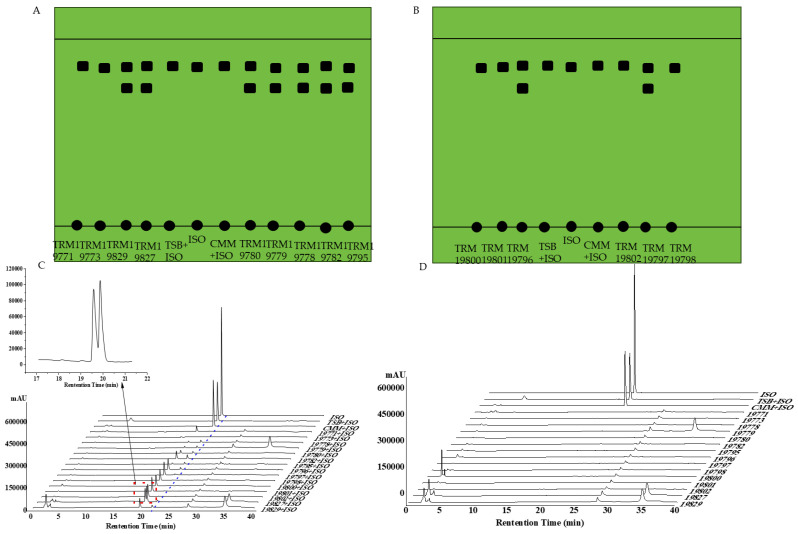
The results of 0 h and 48 h HPLC of bacterial solution mixed with ISO. (**A**) Thin-layer chromatography results of strains. (**B**) Thin-layer chromatography results of culture medium (**C**) 15 strains cultured with isoflavone solution for 48 h. (**D**) Fifteen strains cultured with isoflavone solution for 0 h. Note: Owing to the poor clarity of the original image, a hand-drawn TLC diagram was created using PowerPoint software (Version 2412 Build 16.0.18324.20092).

**Figure 9 sensors-25-00827-f009:**
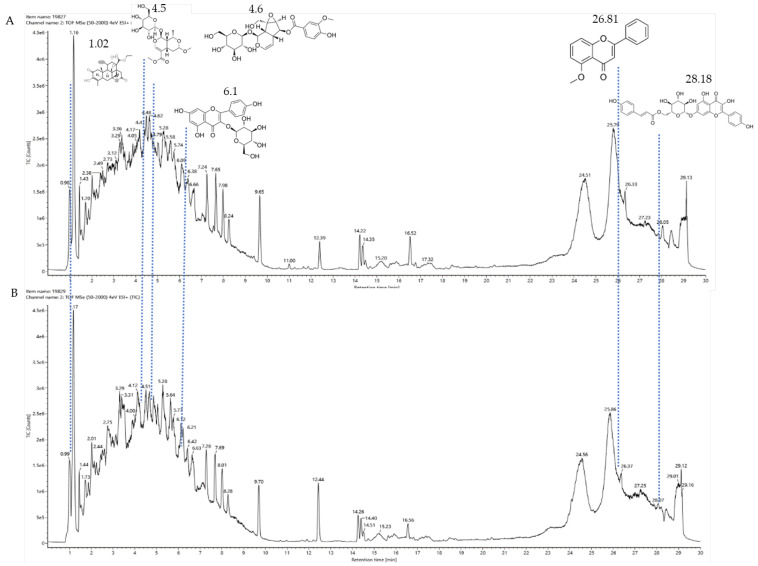
Results of UPLC-MS for strains TRM19827 and TRM19829. (**A**) The results of the liquid chromatography–mass spectrometry analysis of the fermentation broth of strain TRM19827 mixed with isoflavone solution. (**B**) The results of the liquid chromatography–mass spectrometry analysis of the fermentation broth of strain TRM19829 mixed with isoflavone solution. The blue dotted line represents the time and structure of the common compound of the two results.

**Table 1 sensors-25-00827-t001:** The isolation results of different samples and medium strains and the screening results of microplate reader, TLC, and HPLC.

Culture Medium	LB	R_2_A	CMM	ISP4	NA	PDA	MMM	GS 1	TSB	Total
Licorice root	7	2	21	3	3	6	8	1	10	61
Licorice residue	7	7	10	3	4	5	1	4	9	50
Altun Mountain soil sample	0	2	4	0	0	0	1	2	2	11
Total	14	11	35	6	7	11	10	7	21	122
Microplate reader screening strains	2	0	11	0	0	0	2	0	0	15
Thin-layer screening strains	2	0	11	0	0	0	2	0	0	15
Liquid phase screening strains	2	0	11	0	0	0	2	0	0	15
Percentage	14.29%	0.00%	31.43%	0.00%	0.00%	0.00%	20.00%	0.00%	0.00%	12.30%

Note: After isolation from the culture medium, fermentation was carried out using CMM medium and TSB culture medium, and the screening experiment with the microplate reader was conducted using the fermentation broth of the purified strains.

**Table 2 sensors-25-00827-t002:** Study on repeatability, detection limit, and quantitative limit of ISO microplate reader.

Serial Number	1	2	3	4	5	6
OD value	2.583	2.591	2.595	2.591	2.585	2.595
Slope of the standard curve S	0.995					
Standard deviation of the blank sample δ	0.005					
Relative standard deviation (RSD)	0.194%					

**Table 3 sensors-25-00827-t003:** Names and response values of certain compounds following ISO transformation.

Compound Name	Structure	Response	Compound Name	Structure	Response
6-methoxy-2-(2-phenylethyl)-chromone	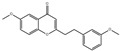	2955	Huangqi Glycoside	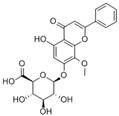	2885
3,5,3′,4′-Tetrahydroxy-6,7-dimethoxyflavone	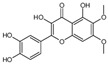	3226	Suhua Flavonoid	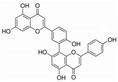	3116
Delphinidin-3-glucoside	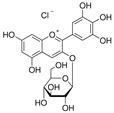	3306	Cyanidin-3-glucoside	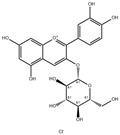	3188
Wangchunhua Flavonol Glycoside A	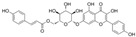	2945	6-Aldose Isomaltol Flavone B	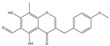	1802
5′-Methoxy Ginkgo Flavone	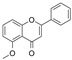	3037	Isoquercitrin-3-O-β-D-Sophora Disaccharide	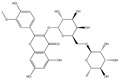	1779
3′,4′,7-Tribenzylsappanol	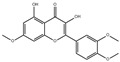	3117			

Note: The unit of response is counts.

## Data Availability

The original contributions presented in this study are included in the article. Further inquiries can be directed to the corresponding authors.

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
