# Peer review of "Microplate Reader–TLC–HPLC–UPLC-MS: A Rapid Screening Strategy for Isoliquiritigenin-Transforming Bacteria"

_sensors, 2025, doi:10.3390/s25030827_

Round 1

Reviewer 1 Report

Comments and Suggestions for Authors

In this submitted article, the Authors present an innovative strategy based on the MTHM (Microplate Reader-TLC-HPLC-UPLC-MS) method for the rapid large-scale screening of isoflavone transformation strains. A significant number of strains (approximately 1500) were investigated through the combination of different techniques (microplate reader, thin-layer chromatography, high-performance liquid chromatography, and mass spectrometry), leading to the identification of fifteen strains with notable transformation capabilities. Furthermore, the study reports the detection of novel bioactive flavonoid compounds.

Overall, this research article is well-written with clearly described experimental methods and well-presented results. Hence, I consider the manuscript suitable for publication in Sensors journal after minor revisions, here listed:

- Please carefully review the manuscript for any typographical errors and punctuation issues (some are highlighted in yellow in the attached file).

- OD is optical density? Please explain the acronym.

- In general, the figures are too small to see clearly the chemical formulas or the written parts. Please increase the size and resolution for improved readability.

- The relative standard deviation reported in Table 2 is different compared to the value reported in the text (line 275), please rectify this discrepancy.

- The TLC plates in Figure 8A-B are currently difficult to interpret. The written annotations are illegible, and the spots are not clearly visible. If obtaining a higher-quality image is not feasible, consider replacing the images with clear hand-drawn representations of the TLC plates to effectively convey the results.

- In Table 3, the units for the reported "response values" are missing. Please specify the unit.

Reviewer 2 Report

Comments and Suggestions for Authors

This manuscript successfully developed an efficient technique for the rapid screening of isoliquiritigenin-transforming strains using the MTHM. The idea is interesting and of originality in this manuscript, but contains more inappropriate or incorrect description in the paper. For these reasons, I do not suggest that this paper is published in this journal now. However, after major revisions, the manuscript may be submitted again.

The inappropriate or incorrect description in the paper including:

1.     The guide to Figure 2 is not found in the text.

2.     The form of the table needs to be reformatted, example table 1.

3.     After the analysis, I would like to ask if in the text, you have written Figure 3 instead of Figure 4.

4. In the explanation of figure 3: (D) The relationship between isoflavone concentration (0.5-5 μg/mL) and R2 at wavelength. The statement here is not accurate.

5.     In section 4, the last sentence in Line 455 “4. Discussion” should not appear.

Round 2

Reviewer 2 Report

Comments and Suggestions for Authors

This manuscript successfully developed an efficient technique for the rapid screening of isoliquiritigenin-transforming strains using the MTHM. The idea is interesting and of originality in this manuscript.

The inappropriate description in the paper including:

1.     In Line 265 “Fig 3C-G” or “Fig 3C-H”?

2.     The figure is too crowded (Fig. 3- Fig. 6).

Comments on the Quality of English Language

 The English could be improved to more clearly express the research.
